# Chromatic covalent organic frameworks enabling in-vivo chemical tomography

Song Wang [1,5], Yangyang Han [1,5], Vaishnavi Amarr Reddy [2], Mervin Chun-Yi Ang [1], Gabriel Sánchez-Velázquez [3], Jolly Madathiparambil Saju [2], Yunteng Cao [4], Duc Thinh Khong [1], Praveen Kumar Jayapal [1], Raju Cheerlavancha [1], Suh In Loh [1], Gajendra Pratap Singh [1], Daisuke Urano [2], Sarojam Rajani [2], Benedetto Marelli [1,4] ✉ & Michael S. Strano [1,3] ✉

Covalent organic frameworks designed as chromatic sensors offer opportunities to probe biological interfaces, particularly when combined with biocompatible matrices. Particularly compelling is the prospect of chemical tomography – or the 3D spatial mapping of chemical detail within the complex environment of living systems. Herein, we demonstrate a chromic Covalent Organic Framework (COF) integrated within silk fibroin (SF) microneedles that probe plant vasculature, sense the alkalization of vascular fluid as a biomarker for drought stress, and provide a 3D in-vivo mapping of chemical gradients using smartphone technology. A series of Schiff base COFs with tunable pKa ranging from 5.6 to 7.6 enable conical, optically transparent SF microneedles with COF coatings of 120 to 950 nm to probe vascular fluid and the surrounding tissues of tobacco and tomato plants. The conical design allows for 3D mapping of the chemical environment (such as pH) at standoff distances from the plant, enabling in-vivo chemical tomography. Chromatic COF sensors of this type will enable multidimensional chemical mapping of previously inaccessible and complex environments.

There has been significant interest in 2D Covalent Organic Frameworks (COFs) as porous organic matrices crafted from the assembly of rigid monomers into consistent crystalline structures[1,2]. Their intricate and adaptable skeletal structures[3] enable precision-tuning of physicochemical properties high photostability[4], expansive surface areas for molecular binding[5], high porosity, and specific nanopore structures[6]. These attributes are particularly advantageous for chemical sensing. By embedding functional groups into the COF skeleton, every unit cell of the COFs can provide an accessible docking site for analyte binding, potentially facilitating fast sensor responses and high sensitivities. Over the past decade, the capabilities of COF-based sensors have been demonstrated in the detection of a wide spectrum of chemical analytes, encompassing nitro explosives[7], metal ions[8] solvents[9], gaseous entities[10], and nucleic acids[11]. Because of limitations in the ability to interface such sensors with biological tissues, examples to date have necessarily been restricted to ex vivo analyses, yielding singular point-based analyte concentration[12].

Here, we show a sensing, colorimetric COF interface can be designed using silk fibroin (SF) microneedles that extend the sensor into the in vivo environment for the rapid detection of physiological proton concentration changes (Fig. 1A). The choice of SF microneedles is based on their inherent robustness, transparency, and their

[1]Disruptive & Sustainable Technologies for Agricultural Precision, Singapore-MIT Alliance for Research and Technology Centre, Singapore 138602, Singapore. [2]Temasek Life Sciences Laboratory Limited, Singapore 117604, Singapore. [3]Department of Chemical Engineering, Massachusetts Institute of Technology, Cambridge, MA 02139, USA. [4]Department of Civil and Environmental Engineering, Massachusetts Institute of Technology, Cambridge, MA 02139, USA. [5]These authors contributed equally: Song Wang, Yangyang Han. ✉e-mail: bmarelli@mit.edu; strano@mit.edu

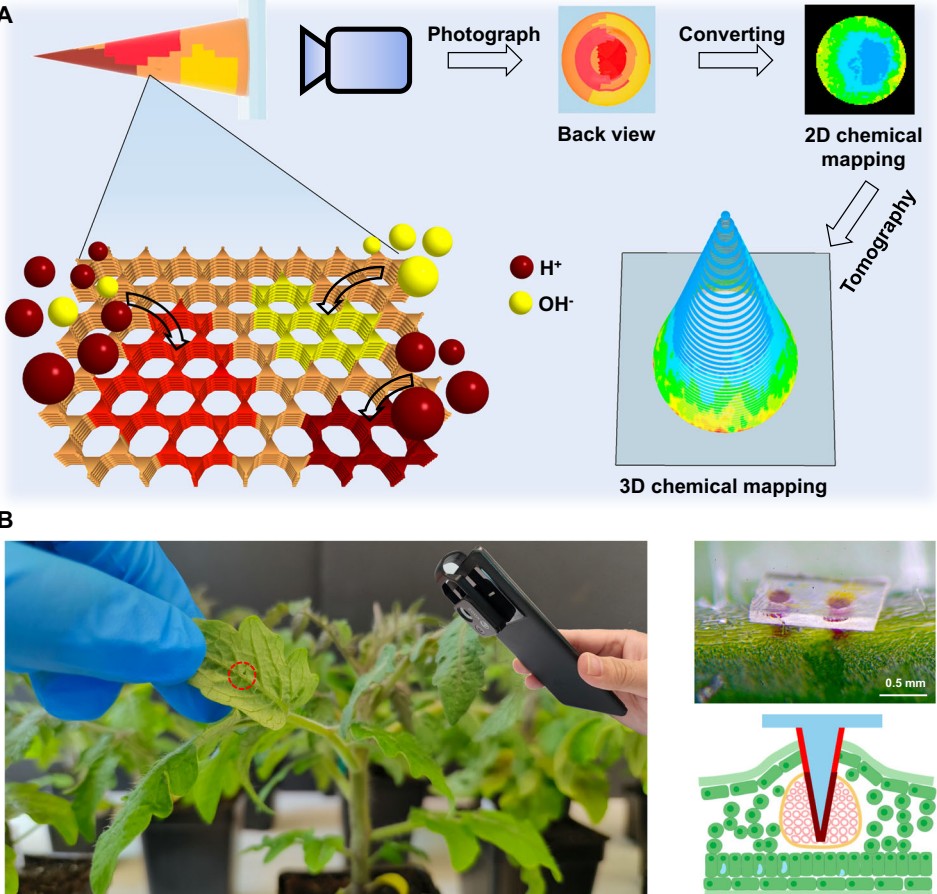

**Fig. 1 | Schematic of the use of chromic covalent organic framework-silk microneedle for microtomographic mapping of physiological chemical. A** The microneedles are designed to reach the bio tissue and the colorimetric COF immobilized on the surface of the microneedles provides a readout of the physiological pH, which can be read with a smartphone camera given the transparency of the silk microneedles. pH distribution in plant tissue can be obtained by converting the color mapping of microneedle to pH mapping, and then to 3D pH mapping via tomography technology. **B** Photos and illustrations (Created in BioRender. Marelli (2023) BioRender.com/p01y968) of the microneedles injected into the leaf midrib of a 6 week old tomato plant for pH monitoring of vascular sap.

capability to remain non-biodegradable upon interfacing with biological matrices[13,14]. The conical microneedle enables a form of chemical tomography, mapping 2D colorimetric data from the COF-SF microneedle tip into the full 3D spatial profile of the specific analyte therein. This can be used to effectively visualize proton concentration gradients as the microneedles penetrate further into the tissues. To validate the applicability of the COF-SF microneedles in providing in vivo chemical tomographic insights, we employed them to map the pH distribution within living tobacco and tomato plants (Fig. 1B). Despite their minimally invasive nature, the COF-SF microneedles can access deeper plant tissues, notably the vascular. Importantly, through COF-enhanced chemical tomography, early pH changes in the xylem under drought stress conditions can be identified, offering a detection advantage over traditional commercial biomarkers. Such early detection is pivotal, as it offers the potential for proactive intervention and pre-symptomatic diagnosis against drought stress in agricultural settings before irreversible yield loss occurs.

## Results and discussion

### Rational design of the pH-sensitive imine COF series

COFs are emerging candidates for biosensor materials. Recent advances have shown their succussed in biosensing with targets of DNA[11], antibiotics[15], as well as biomarkers[16], showing the promise for selective detection of physiological molecules of plant. The pH sensitivity of COFs we demonstrate in this work, can be engineered by

structural modifications[17–20]. Since most biological tissues display pH levels from near-neutral to mildly acidic or alkaline[21], COFs that visibly shift color in these conditions suit in vivo sensing. The abundant nanopores of crystalline COFs can dramatically decrease the diffusion resistance of guest molecule, thus enable the ultrafast chromic response. And the COF is insoluble 2D polymer, that can essentially avoid leakage of small molecular dye and plant tissue contamination. Adjusting the electron donor/acceptor properties of COF monomers can fine-tune their proton affinities and electron delocalization within the framework. Accordingly, we employed three amine monomers, i.e., 2,4,6-Tris(4-aminophenyl)pyridine (TAPP), 1,3,5-tris (4-aminophenyl) benzene (TAPB), tris(4-aminophenyl)amine (TAPA); and three aldehyde monomers, i.e., 2,5-Dimethoxyterephthalaldehyde (DMTA), Tris(4-formylphenyl)amine (TFPA), 2,4,6-Tris(4-formylphenyl)pyridine (TFPP), to develop a series of COFs: TAPP-DMTA, TAPP-TFPA, TAPB-TFPA, and TAPA-TFPP (Fig. 2A and Supplementary Fig. S1), to study the effects of electron affinity[22] and the role of constitutional isomerism[23] on proton dynamics. Imine will be the first protonation point when it existed together with pyridine or triphenylamine[24,25]. Polymerization was visible from the disappearance of amine[26] and aldehyde[27] signals and the rise of imine bonds[28] seen through attenuated total reflectance-Fourier transform infrared spectroscopy (ATR-FTIR, Supplementary Fig. S2), while X-ray Diffraction (XRD) analyses verified the crystallinity (Supplementary Figs. S3–6). And the crystallinity of COF is robust enough that can maintain well even after soaking in aqueous

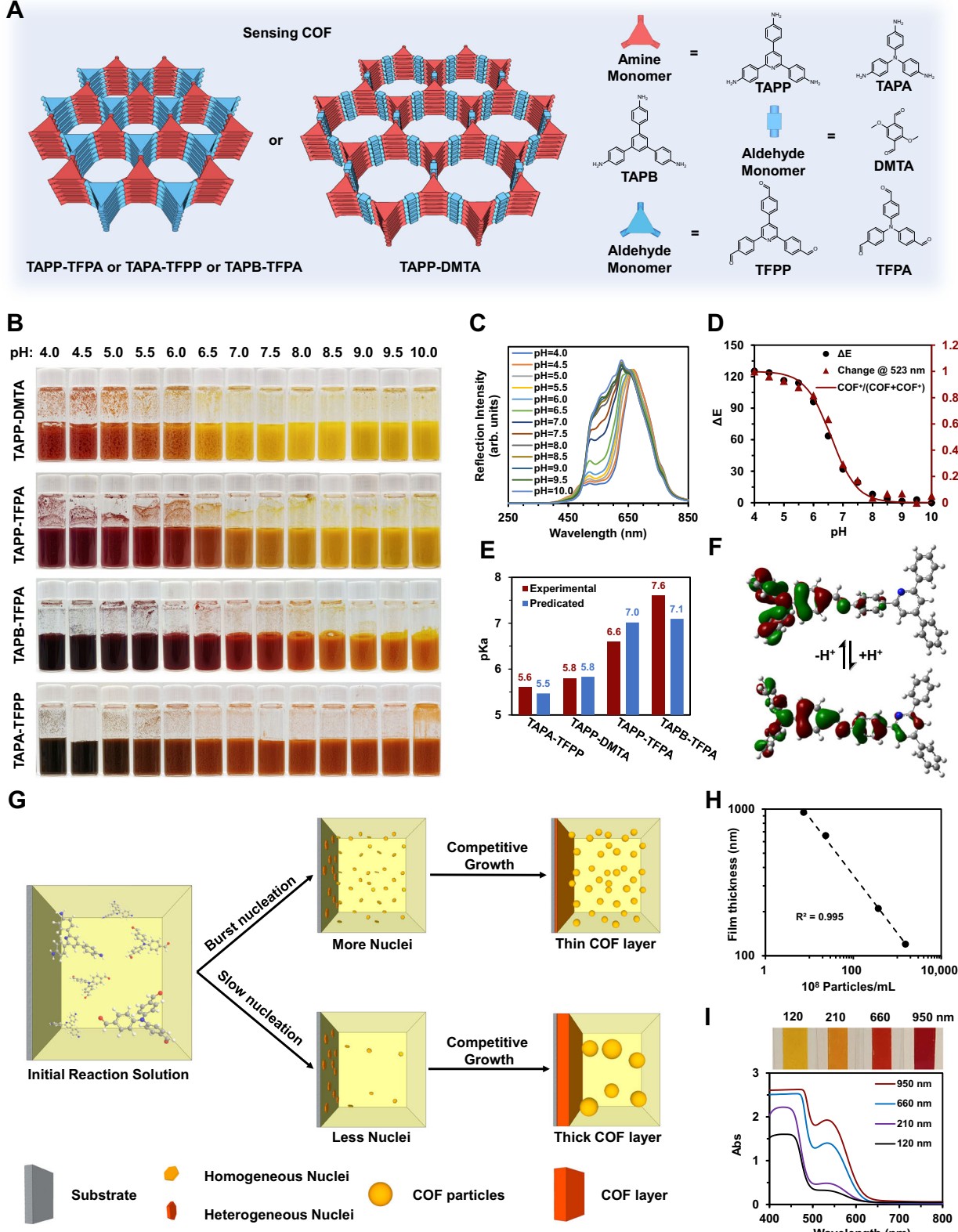

Fig. 2 | Chromatic COF structures and their thickness control. A Structures of TAPP-TFPA, TAPA-TFPA, TAPB-TFPP and TAPP-DMTA and their monomers. B Photos of TAPP-DMTA, TAPP-TFPA, TAPB-TFPA and TAPA-TFPP powders dispersed in buffers at different pH values. C Reflection spectrum of TAPP-TFPA powders treated by different pH buffers. D ΔE and relative reflection intensity change at 523 nm of TAPP-TFPA powder varies with pH value and relative pKa = 6.6 curve (fitted according to Henderson-Hasselbalch equation). E Experimental and predicated pKa of different COFs. Predicated pKa is obtained according to simulated proton affinity in water medium. F Density functional theory (DFT) optimized molecular structures and highest occupied molecular orbitals (HOMOs) of unprotonated and protonated TAPP-TFPA repeat unit. G Illustrations of homogeneous nucleation limitation strategy. H Relationship of film thickness and particles density for TAPP-TFPA. I Photos and UV-Vis absorption spectra of TAPP-TFPA films with different thickness on PET substrates treated with pH 5.0 buffer.

buffer solutions with pH values from 4 to 10 for one day (Supplementary Fig. S7).

To assess the correlation between proton binding and chromatic shifts, each COF was dispersed in buffer solutions with pH levels spanning 4.0–10.0 (Fig. 2B). All four imine-based COFs, being Lewis bases, showcase acid chromism: their colors transition from yellow to red or from orange to dark red as the pH decreases. The difference in reflection intensity at 523 nm (Fig. 2C) and its value at pH 10.0 as reference can be related to the protonation state of the COF assuming a single proton binding equilibrium (Fig. 2D and Supplementary Fig. S8).

$$\Delta E = \frac{[COF^+]}{[COF^+] + [COF]} = \frac{1}{1 + 10^{(\text{pH} - pKa)}} \qquad (1)$$

Here, [COF] and [COF$^+$] are concentrations of unprotonated and protonated sites respectively. The differential energy (ΔE) of TAPP-TFPA was observed to fluctuate significantly (14-fold) within the pH range of 5.5–8.0, with minimal deviations outside this window, mirroring the trends seen with reflection intensity shifts at 523 nm. The other synthesized COFs demonstrated analogous pH dependencies, with pKa values being contingent on their inherent alkalinity, demonstrating the ability to control the sensitivity range of the chromatic sensor. This interdependence aligns well with the Henderson-Hasselbalch equation (Eq. (1), Fig. 2D and Supplementary Fig. S8)[29]. A composite of the acceptor amine monomer TAPP and the donor aldehyde monomer DMTA, TAPP-DMTA, yielded a pKa of 5.8 (Fig. 2E and Supplementary Fig. S8). Substituting DMTA with the potent donor aldehyde monomer, TFPA, escalated the pKa to 6.6 for TAPP-TFPA (Fig. 2D, E). Further monomer replacement, introducing the weaker donor TAPB to get TAPB-TFPA, rendered the highest pKa value of 7.6 (Fig. 2E and Supplementary Fig. S8). These results underline the ability of the monomer's electron donor potential in adjusting the pKa values of the COFs. Interestingly, the constitutional isomerism of the imine linkage orientation seemed to decrease the pKa, evidenced by the isomer TAPA-TFPP having a reduced pKa of 5.6 compared to TAPP-TFPA. Such isomer also displaying a distinct chromatic range (Fig. 2B). Importantly, the acidichromism performance has good reversibility, that was maintained after switching between pH 4 and pH 10 buffer solutions for 10 times (Supplementary Fig. S9). The acidichromism is also irrelevant to the ion concentrations or the present of phytohormones (Supplementary Fig. S10).

Density functional theory (DFT) simulations on the COF repeat units were employed to investigate their pH-sensitive chromatic behavior (Fig. 2F and Supplementary Fig. S8). The highest occupied molecular orbitals (HOMOs) of the unprotonated form of TAPP-TFPA are predominantly located within the TFPA moieties. Post protonation, there is a relative delocalization of the HOMO orbitals across the framework. A similar delocalization pattern was noted in the other COFs upon protonation (Supplementary Fig. S11), elucidating the red shifts witnessed with pH reductions. Proton affinities in aqueous environments were calculated as energy disparities between protonated and deprotonated repeat units (Fig. 2F). Predicted pKa values based on the proton affinities in water[30] were determined to be 5.5, 5.8, 7.0, and 7.1 respectively for TAPA-TFPP, TAPP-DMTA, TAPP-TFPA, and TAPB-TFPA, reaffirming the pKa findings showcased in Fig. 2E.

**An interface for chemical tomography**
We selected TAPP-TFPA for sensor development due to its distinct color transition within the physiologically relevant pH range. However, the synthetic results in the previous section, along with the success of DFT predictions of the pKa, highlight the newfound capability to tailor the sensitivity range for sensors of this form.

The sensing COF in film form displays thickness-dependent photoabsorption and reflectance that vary with wavelength, allowing

transmission rather than absorption of longer wavelengths in films[31] under 600 nm. Despite substrates promoting even COF layers from synthetic solutions (Fig. 2G), films typically remain under 500 nm due to competitive growth between heterogeneous and homogeneous nuclei[17,32–35], causing a blue shift and altering the pH detection range. To address this, we introduced a homogeneous nucleation limitation (HNL) approach to produce COF layers with thicknesses greater than 500 nm (Fig. 2G and Method of SI). Restricting burst homogeneous nucleation in the solution allows more COF material to bind to the heterogeneous nucleation sites on substrates, producing a thicker COF film. By reducing the nuclei density from $1.5 \times 10^{11}$ /mL to $2.0 \times 10^8$ /mL, COF film thickness increased from 120, 210, 660, to 950 nm (Supplementary Figs. S12–16). This relationship between nuclei density and thickness was found to be linear in a logarithmic plot (Fig. 2H). The surface morphology and roughness (Ra= 18.3–26.2 nm) of these films were confirmed to be unaffected through both AFM and SEM analyses (Supplementary Figs. S16 and 17). Interestingly, the crystallinity of COF also improved by reducing the homogeneous nuclei density (Supplementary Fig. S18). We attributed this to the better reversibility of the COF synthetic system[17].

Upon exposure to a pH = 5.0 acidic buffer, the TAPP-TFPA film at 120 nm thickness exhibits a yellow hue, diverging from its intrinsic red at the same pH (Fig. 2B). At thicknesses of 210, 660, and 950 nm, the film shifts to orange, bright red, and scarlet. CIELAB analysis for these films (Supplementary Fig. S19) shows a positive correlation between the a* value and thickness. At a 950 nm thickness, it's a* value aligns with that of the TAPP-TFPA powder. This convergence suggests enhanced absorption efficiency for longer wavelengths, as evidenced by the increased absorption peak at 533 nm in the UV-Vis spectra of thicker films (Fig. 2I). Therefore, our subsequent experiments focused on the TAPP-TFPA COF film with a thickness of 950 nm. The TAPP-TFPA films exhibited distinct color changes in response to buffers with varying pH levels (Supplementary Fig. S20), and a pKa of 5.6 was determined by analyzing the absorption change at 533 nm using the Henderson-Hasselbalch equation (Supplementary Fig. S21).

Leveraging the HNL technique, we successfully fabricated COF-SF microneedles, designed to access deeper plant tissues, a feat challenging for conventional spectroscopic tools[13,14,36]. Soft replica molding yielded the SF microneedle arrays[14], which were then coated with the TAPP-TFPA layer, directed by the HNL strategy (Supplementary Fig. S22). A sonication process in ethanol selectively eliminated the TAPP-TFPA layer on the microneedles' bodies, retaining only the tip layers (Supplementary Figs. S23 and 24). The resultant TAPP-TFPA SF microneedles (TSMNs) upheld their original form, exhibiting an orange hue post-ethanol drying (Fig. 3A). A SEM image of a fractured TSMN tip confirmed the TAPP-TFPA layer thickness to be roughly 1 μm (Fig. 3B). Various molds facilitated the modulation of microneedle sizes and shapes, allowing to generate needles with heights of 250, 450, 700 μm, named as TSMN250, TSMN450 and TSMN700, respectively (Fig. 3C).

The transparency of SF microneedles allows for in situ observation of the COF layer's pH-sensitive chromatic response from the top view of the TSMNs. If the TSMN tips alter color due to pH changes, the relative height of the color-altered segments (h/H) when viewed laterally should match the relative diameter (d/D) when observed from the top (Fig. 3D). Validation of this hypothesis was achieved by immersing the TSMN700 tip into a transparent alkaline gel (pH = 9.0) and observing the resultant top view (Fig. 3E). The relationships between d/D and h/H were found to match the geometric relationship of d/D = h/H (Fig. 3F). This suggests that in situ monitoring of pH spatial distribution in plant tissues accessed by TSMNs is feasible by analyzing the color transitions at the microneedle tips. According to the geometry of the microneedle, the viewing angle deviation of top view need to be minimized, and kept within -9° to get all the pH spatial information of the microneedle (Supplementary Fig. S25). The TSMN

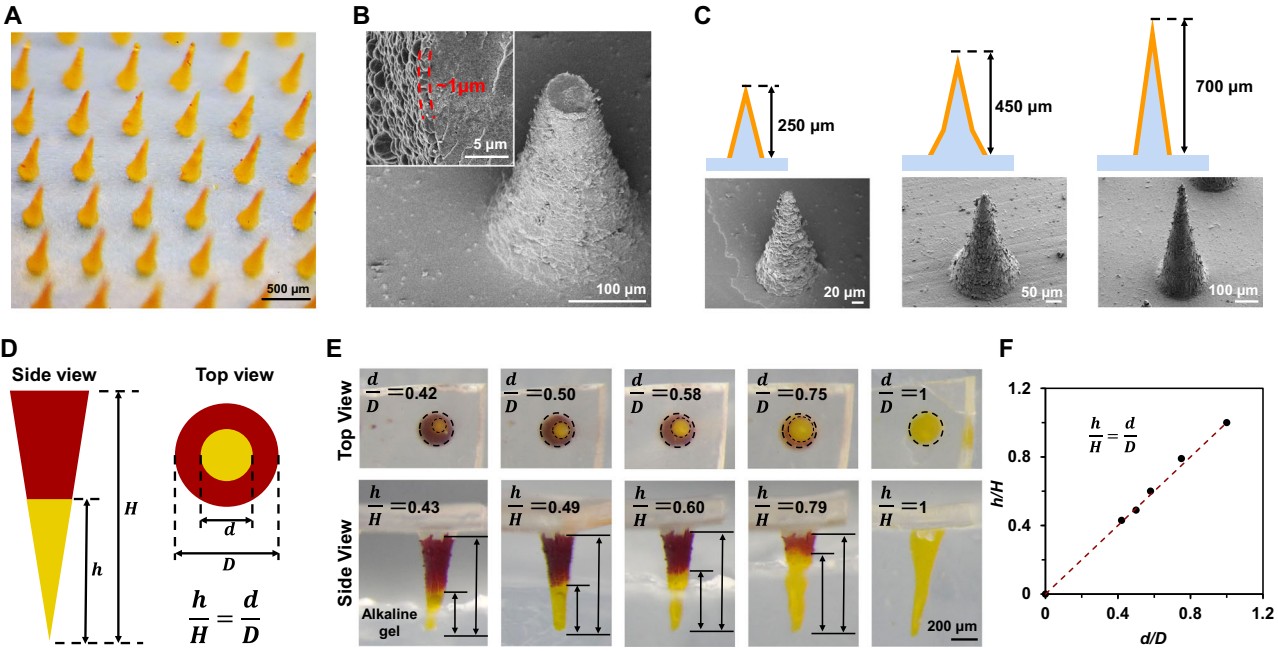

**Fig. 3 | Fabrication and characterizations of COF-silk microneedles. A** Optical microscope image of TSMN700. **B** SEM images of tip-broken TSMN700 that depict the TAPP-TFPA outer layer. **C** Illustrations and SEM images of TSMN250, TSMN450, and TSMN700. **D** Relative height (defined as the ratio between the length of the segment that depicted a color change ($h$) and total length ($H$)) is equal to the relative diameter (defined as the ratio between the diameter of the color changed inner circle ($d$) and the total diameter ($D$)) observed from a top view. (**E**) Top and side views of TSMN700 inserted into a transparent alkaline gel. TSMN700 was pretreated with pH = 4.5 buffer. Alkaline gel has a pH of 9.0. **F** Relative diameter and relative height of color changed microneedle part presents a linear relationship. In **A–C**, **E** images are representative of at least three independent experiments.

exhibited a pKa value of 6.0 (Supplementary Fig. S26) and a pH chromic response time of 8 s (Supplementary Fig. S27 and Supplementary Movie S1).

## In vivo chemical tomography

This unique sensing platform offers a gateway into in vivo chemical tomography, enabling detailed spatial mapping of pH distributions within complex, living tissue samples. To showcase the effectiveness and applicability of this tomography, we conducted experiments on leaves of two dicotyledonous species: *Nicotiana tabacum L* (tobacco) and *Lycopersicon esculentum* (tomato). Given the vascular architecture of these plants, where xylem vessels are arranged in a ring formation in stems and petioles and are densely packed within leaf veins, our microneedles can strategically and consistently access vascular-rich areas[37–39]. This vascular topology facilitates the TSMNs' capacity to gauge the pH of the vascular sap, particularly when inserted at the midrib of dicotyledonous leaves (Fig. 1A, B). Mechanical testing revealed that the TSMN700 microneedle withstands a compressive force of up to 0.13 N before displacing 100 μm–exceeding the 0.045 ± 0.018 N required for midrib penetration–and pierces tobacco leaf veins without deforming, with post-injection histology confirming successful vasculature access (Fig. 4A, B, Supplementary S28). Importantly, even after such an insertion, we observed no significant deviations in physiological markers like NBI (Chlorophyll content/flavonol content) when comparing injected leaves to controls after a 15-day period (Fig. 4C, Supplementary S29), despite a 300 μm wound evident post-removal (Fig. 4B). This is also supported by the chlorophyll fluorescence imaging analysis, showing high photochemical efficiency ($F_V/F_M$ and ΦPSII) as well as non-photochemical quenching (NPQ) after the injection for 1 h and 24 h (Supplementary Fig. S30). We also employed rhodamine B to monitored the water transport across the petiole and midrib, which evidences that the needles do not

obstruct water transport in the plant vascular system (Supplementary Fig. S31).

We investigated the capabilities of the sensing microneedles to accurately detect pH alterations in the vascular (mainly xylem) fluid, whether transpired from the roots or introduced via petiole injections. Using alkaline and acidic buffers as test agents, upon microneedle insertion, the sensing domain established contact with the vascular bundle. Simultaneously, its deeper section penetrated further, accessing the epidermal or mesophyll tissues (Fig. 4D). This interaction resulted in spatially dependent color modifications, effectively enabling the microneedle to create a spatial pH-map of the living tissue in real-time. For instance, Fig. 4E displays the pH variation in a freshly excised leaf post-petiole immersion in an alkaline solution with pH 9.5. Within 30 min, the pH rose from 5.5 to 8.5. Switching the leaf to an acidic medium (pH 4.5) resulted in a pH stabilization at 5.0 after the same time. Significantly, the microneedle's penetration depth–and by extension, the depth of sensing–can be modulated by altering its length (Fig. 4F). At 450 μm, the pH shifted from 5.5 to 6.5 within 10 min. In contrast, at 250 μm, the pH stayed constant after 30 min, underscoring the microneedle's ability to access specific tissue layers within the living leaf.

Complementary TSMN-based pH analysis indicates stable pH in plant tissues like the epidermis and mesophyll after exposure to alkaline or acidic sap within the tested period. This consistency leads us to propose the TSMN700's outer circle colors as a reliable reference, with the inner circle acting as a dynamic pH indicator. To demonstrate the TSMN700's sensitivity in relation to vascular pH, we exposed entire tobacco roots to a mild alkaline buffer (pH 8.0) (Fig. 4G). Following midrib injection, the inner circle's pH rose to near 7 within an hour and stabilized by hour two. Subsequent root immersion in an acidic solution (pH 4.5) decreased the inner circle's pH to about 6 initially, leveling off at around 5 by the second hour.

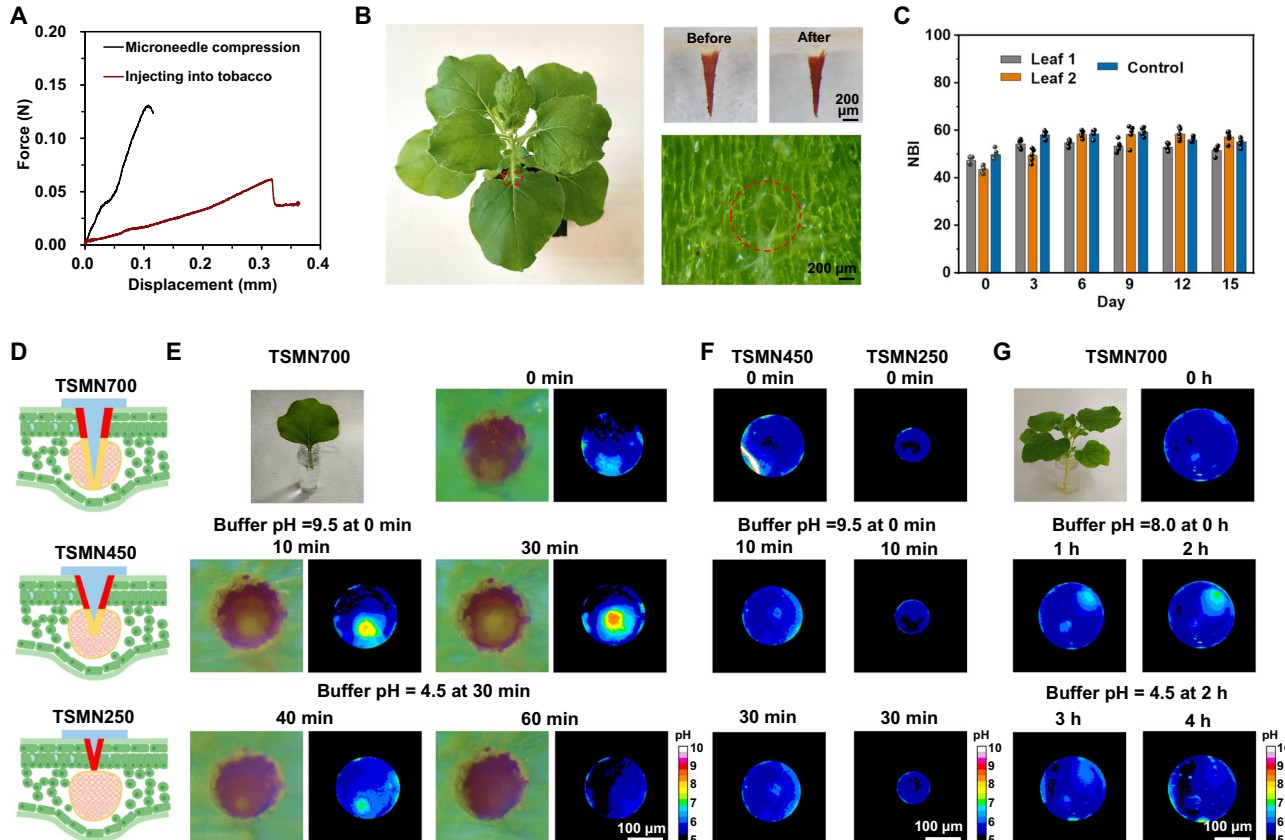

**Fig. 4 | In situ detection of exogenous pH change in plants by microneedles.**
**A** Mechanical behavior of single TSMN700 microneedle under compressive force and its penetration into a tobacco midrib under compressive load. **B** Digital photo (left) of a tobacco leaf after injecting two TSMN700 microneedles at the end of vein for 4 days. TSMN700 microneedle before and after the injection (top right) and optical microscope image of tobacco vein after removing the TSMN700 (bottom right). **C** Nitrogen balance index (NBI) measured in tobacco leaves injected by a TSMN700 to monitor its health while carrying the microneedle up to day 15. Five different leaf points were measured. **D** Illustrations (Created in BioRender. Marelli (2023) BioRender.com/p01y968) of color change of TSMN700, TSMN450 and TSMN250 in the midrib of a tobacco leaf after absorbing alkaline buffer. **E** Digital

photo (top left) and optical microscope images from the top view of TSMN700 injected in the midrib of tobacco leaf, and relative pH mapping images obtained according to the calibration of TAPP-TFPA in different pH buffer. The leaf was cut off and its petiole was submerged into pH = 9.5 buffer at 0 min and pH = 4.5 buffer at 30 min subsequently. **F** pH mappings obtained from optical microscope images of TSMN450 and TSMN250 injected in the midrib of tobacco leaves. Leaves were cut and their petioles were submerged into pH = 9.5 buffer at 0 min. **G** Digital photo (top left) and pH mappings obtained from optical microscope images of TSMN700 injected in the midrib of tobacco leaves. The whole plant was kept intact and its roots were first submerged into pH = 8.0 buffer and moved in a pH = 4.5 buffer after 2 h. In **B**, **E−G** images are representative of three independent experiments.

---

We attribute this delayed response in pH adjustments to the extended transportation pathways and inherent biological barriers presented by the roots.

## Tomographic chemical imaging of tomato plants undergoing drought stress

Under conditions of water scarcity, plant roots independently initiate long-distance chemical signaling to the plant canopy, modulating leaf growth and stomatal conductance. This mechanism is associated with an inhibition of the root plasmalemma P-type H + -ATPase, resulting in a discernible shift in the pH of xylem sap from roots to the canopy at the onset of drought stress[40,41].

To validate the COF microneedle's ability to provide in vivo chemical tomographic imaging of this early stress signal, plants at 6−8 weeks had regular watering halted on day 1, leading to a significant decrease in soil water content over the subsequent 6 days (Fig. 5A). Measurements of flavonoids (Fig. 5B) and Nitrogen Balance Index (NBI, Fig. 5C), which are used as drought stress biomarkers[42–44], showed $p > 0.05$ and $p < 0.0001$ on days 6, respectively, indicating the establishment of a drought-stressed state after day 6. This was supported by the observed yellowing at day 6 (Fig. 5D), both phenotypical indicators of drought stress used in the field.

COF-enabled tomographic imaging detected the early onset of drought conditions by utilizing distinct pH gradients orthogonal to the leaf lamina. TSMN700 microneedles, inserted into tomato mid-ribs abaxially, terminated at the vasculature bundles, as confirmed by histology in Fig. 5E. Fresh TSMN700 microneedles were injected into the mid rid of tomato leaves every day before testing, to prevent the loosen of them with plant tissues after long time. Colorimetric differences of TSMN700 microneedles were observed using a smartphone with a 200X lens (Fig. 5F). The microneedle's conical geometry translates the in vivo 3D gradient at the COF sensing surface to a 2D plane at the leaf lamina, enabling real-time imaging by observing the microneedle's base. Colorimetric changes were translated to pH, with the outer perimeter, which is outside of the vasculature bundle, serving as a reference to normalize the color spectrum across camera configurations. Upon insertion into adequately hydrated tomato plants on day 1, the base of the COF-microneedle exhibited a predominant scarlet hue, suggestive of an interaction with vascular sap exhibiting a pH range of 4.5–5.0. The external perimeter displayed a pronounced red at the upper limit of pH 5.0. Notably, the central region of the base, aligning with the microneedle's most penetrative segment, demonstrated an increasing intensity of hue from day 4 onwards, signifying an alkalinization

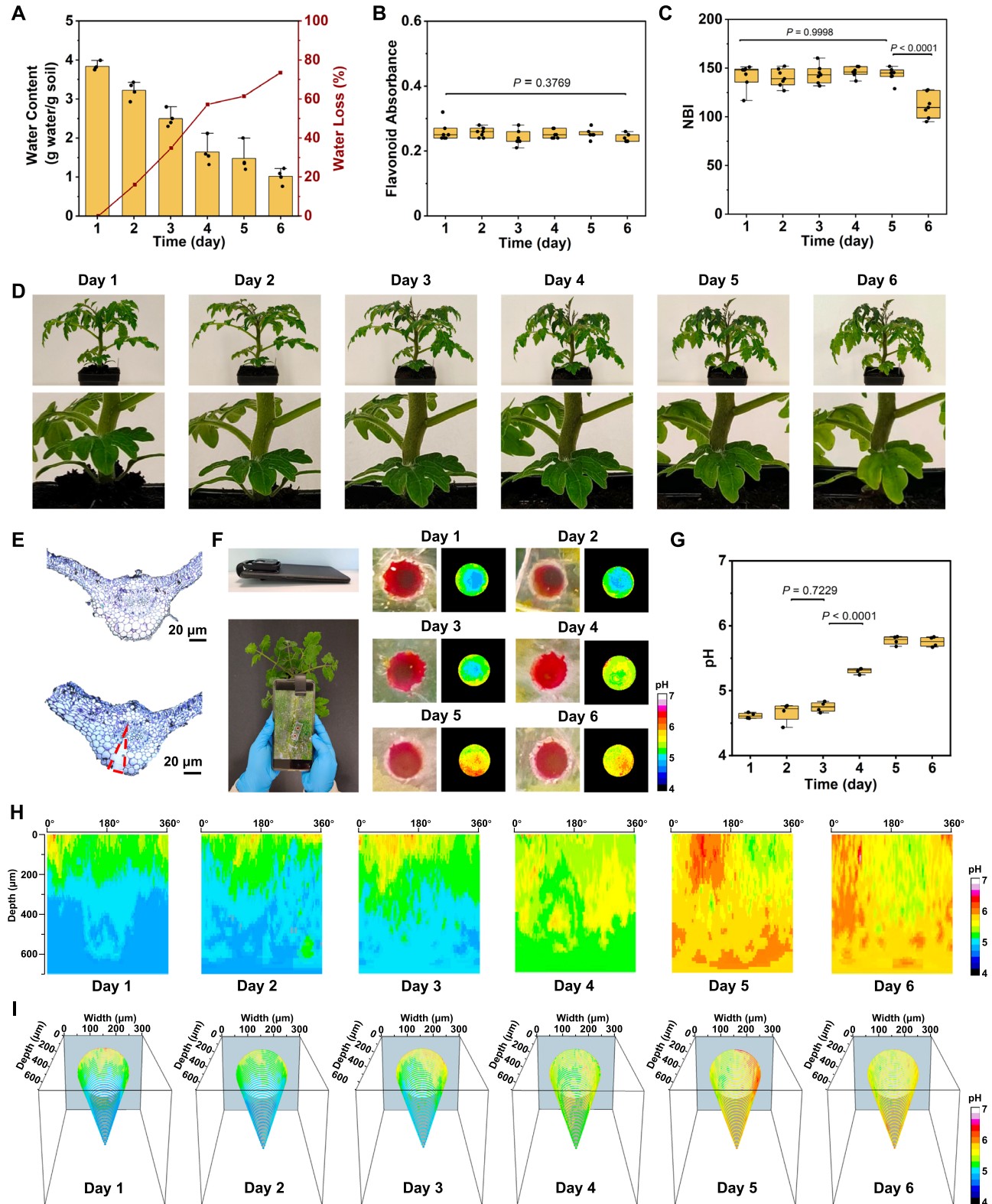

of the vascular sap (Fig. 5F). This gradual shift in basicity of the vascular sap was quantified by averaging pH values from the bases of four distinct microneedles daily, revealing a direct correlation with the initiation of the drought condition (Fig. 5G).

The COF sensing microneedle design facilitates the transformation of the observed 2D pH gradients at its base, $pH_{2D}(\theta,r)$ into a 3D spatial mapping $pH_{3D}(\theta,z)$, encompassing the annular direction ($\theta$) and depth ($z$) within the tissue. This transformation employs the geometric relation:

$$pH(\theta, z) = pH_{3D}(\theta, z) = pH_{2D}\left(\theta, \mathbf{r} = \frac{d_{base}}{2\,z_{max}}\,z\right) \qquad (2)$$

where $d_{base}$ represents the microneedle base diameter in the form of the observation spot, and $z_{max}$ corresponds to the extent of the microneedle's penetration into the tissue. Depth-angle tomographic

**Fig. 5 | Tomographic chemical imaging of living tomato plants undergoing drought stress. A** Water content of soil (Data are means ± SD from four individual samples). At day 1, regular watering was halted causing a decay in soil water content over 6 days. **B**−**D** Stress markers of including flavonoid absorption (**B**) did not show difference during the 6 days ($n = 7$). And NBI (**C**) show deflections starting on day 6, confirming the induction of a water stress state ($n = 7$). Data are means ± SD from seven individual leaf tests. Photographic evidence (**D**) also corroborates the water stress state at day 6 with slightly yellowing. **E** The sensing COF microneedle interface is imaged in the tomato leaf midrib stained with toluidine blue showing before (top) and after (down) injection at the abaxial side. The blue circle indicates the vasculature zone and dashed red triangle indicates the point of microneedle injection. Images are representative of three independent experiments. **F** Data

collection from the TSMN700 sensor injected into the midrib of a tomato leaf was from a smartphone camera at periodic time points after day 1. **G** pH of vascular saps measured by inner circle color difference against outer circle for TSNM700 injected into the midrib of tomato leaves at different time points, after day 1. Data are means ± SD from four individual leaf tests. **H** In-Vivo chemical tomographic images in the form of depth-orientational angle pH maps. **I** Alternative depiction of maps onto the 3D conical profile of the COF microneedle sensor interface. In **B**, **C**, and **G** data are shown as boxplots of the number of indicated samples displaying the maximum and minimum, first and third quantiles, and the median. In **B**, **C**, and **G** data are analyzed by one-way ANOVA followed by the Tukey test and the $P$ values were shown. Source data are provided as a Source Data file.

representations (Fig. 5H) can further be delineated as a 3D conical structure (Fig. 5I), with both visualizations depicting the temporal evolution of tissue acidity in three dimensions. These visual renderings consistently illustrate reduced acidity at the midrib's surface, and the initial acidic pH gradient along the microneedle's extent prior to drought-induced alterations. At day 4, the soil lost 57% of water, which was considered as the starting point for drought conditions[45]. Tomographic analysis identified the beginning of acidity reduction at the same day−a signal of drought stress. These findings underscore that water stress indicators become discernible 48 h prior to detection via prevailing portable optical-based methodologies in plant science and agronomy, and potentially before the detrimental impacts on yield become irreversible. Consequently, these results advocate for the potential of chemical tomography in forecasting plant physiological changes in agricultural contexts, providing a robust method for early detection of stress-associated biomarkers.

We present a chromatic COF-SF microneedle interface that enables in vivo chemical tomography. As an example, we show the real-time spatial mapping of physiological pH in remote plant tissues−a domain traditionally beyond the reach of existing analysis methods. Through a systematic approach involving the design of chromic COF structures and a controlled coating strategy, the COF-SF microneedles were successfully fabricated. These microneedles exhibit the capability to accurately detect and convey in vivo physiological pH variations, delivering detailed spatial distribution data. The inherent colorimetric change of the COF can be visualized using an optical microscope or a smartphone camera. Such observations can be transformed into comprehensive tomographic depth-angle pH mappings and elaborate 3D pH visualizations. Furthermore, the practical application of COF-SF microneedles was validated by their ability to promptly identify the alkalization phenomenon in tomato vascular sap under the duress of drought conditions, providing positive feedback two days earlier than commercial methods. Together, these results constitute a pioneering demonstration of the potential of microneedle-based tomographic chemical imaging to derive biochemical insights from living tissues. This study sets the foundation for the subsequent advancement of COF-SF microneedle-based in vivo sensors tailored for a spectrum of biologically-relevant analytes, thereby equipping researchers with the tools to explore and elucidate the mechanisms underlying key biological processes within different living systems.

## Methods
### COF characterization
COF powders and microneedles were characterized using a Hitachi SU-8010 Field Emission Scanning Electron Microscope (FESEM) at an accelerating voltage of 3 kV. To enhance conductivity during imaging, samples were subjected to a Pt coating for 60 s. Absorption spectra of COF films deposited on PET substrates were obtained using an Agilent Cary 5000 UV-Vis spectrophotometer, with a pure PET substrate in buffer serving as the background reference. The chemical composition of COF and monomer powders was further analyzed using Attenuated

Total Reflectance Fourier Transform Infrared (ATR FT-IR) Spectroscopy on a VERTEX 70 FTIR spectrometer from Bruker. To assess the crystalline nature of COF powders, X-ray Diffraction (XRD) measurements were conducted using a PANalytical X-pert Powder diffractometer with a Cu anode, a wavelength of 1.5418 Å, a scan step of 0.0262606°, and a step time of 97.983 s. Surface morphology of TAPP-TFPA COF films on PET was examined using Atomic Force Microscopy (AFM) on a Bruker Dimension ICON system with a PPP-NCHR-W probe from Nanosensors™. Before AFM imaging, samples were conditioned in a vacuum oven at 30 °C overnight, and the resulting images were analyzed using NanoScopeAnalysis 2.0 software. Regarding microscopy images, they were all captured on an AmScope −4TP system, utilizing 10× eyepieces, a 2× objective, and a 144-LED ring-light for illumination. Chlorophyll fluorescence imaging analysis was conducted by FluorCam 1300 from Photon Systems Instruments (PSI) company.

### Compression test
Force-displacement profiles were recorded on an Instron 5848 Microtester at a crosshead speed of 0.1 mm/min. For leaf injection force measurements, a pristine microneedle was affixed to a silicon wafer using epoxy. The target leaf was adhered to the upper compression module, while the wafer-mounted microneedle was positioned on the lower module. Careful alignment ensured precise injection into the leaf vein.

### COF powder preparation
The preparation of various COF powders involved dissolving the respective monomers and competitors in a 30 mL solution of mesitylene/dioxane (v/v, 1/4) within a glass media bottle. Amounts of monomers and competitors for different COFs are summarized in the Table 1. To initiate polymerization, 1.5 mL of 1% (v/v) trifluoroacetic acid in mesitylene/dioxane (v/v, 1/4) was added to the solution. The reaction was allowed to proceed at room temperature for one day. Subsequently, the resulting precipitate was filtered using a 0.22 μm polytetrafluoroethylene (PTFE) filter and washed three times with mesitylene/dioxane and three times with ethanol. The wet powder was then dried in a vacuum oven at 30 °C for one day. Yields of products: TAPP-DMTA 82.4 mg (31.0%); TAPP-TFPA 92.3 mg (32.7%); TAPB-TFPA 104.5 mg (37.1%); TAPA-TFPP 64.4 mg (22.8%).

### Preparation of TAPP-TFPA films on PET
To fabricate TAPP-TFPA films, 158.6 mg (0.450 mmol) of TAPP and 148.2 mg (0.450 mmol) of TFPA were dissolved in 30 mL of a mesitylene/dioxane mixture (v/v, 1/4) in a glass media bottle. The thickness of the COF film was regulated by varying the amounts of competitors from 3 to 12 equivalent, with the example provided here using 12 equivalents. Specifically, 546.6 μL (5.40 mmol) of benzaldehyde and 489.6 μL (5.40 mmol) of aniline were added as competitors. A PET substrate was then immersed in this solution. The polymerization was initiated by adding 1.5 mL of 1% (v/v) trifluoroacetic acid in mesitylene/dioxane (v/v, 1/4) to the mixture. The reaction was allowed to proceed

**Table 1 | COF powder synthesis monomers and competitors**

| Powder type | Amino monomer | Aldehyde monomer | Benzaldehyde | Aniline |
|---|---|---|---|---|
| TAPP-DMTA | 158.6 mg (0.450 mmol) | 131.0 mg (0.675 mmol) | 273.2 µL (2.70 mmol) | 244.8 µL (2.70 mmol) |
| TAPP-TFPA | 158.6 mg (0.450 mmol) | 148.2 mg (0.450 mmol) | 546.5 µL (5.40 mmol) | 489.6 µL (5.40 mmol) |
| TAPB-TFPA | 157.8 mg (0.450 mmol) | 148.2 mg (0.450 mmol) | 546.5 µL (5.40 mmol) | 489.6 µL (5.40 mmol) |
| TAPA-TFPP | 130.7 mg (0.450 mmol) | 176.1 mg (0.450 mmol) | 1366.2 µL (13.50 mmol) | 1224.0 µL (13.50 mmol) |

at room temperature for one day. Following this, the glass bottle was sonicated in a bath to detach any powder from the film surface. The film-coated PET was subsequently rinsed three times with mesitylene/dioxane and another three times with ethanol to ensure thorough cleaning. Finally, the TAPP-TFPA film on the PET substrate was air-dried in a fume hood for one hour.

## Preparation of regenerated silk microneedles
Bombyx mori cocoons were cut and degummed in a boiling 0.02 M sodium carbonate solution for 30 min. The cocoons were then thoroughly rinsed with Milli-Q water and allowed to air-dry at room temperature. The dried silk fibers were subsequently dissolved in a 9.3 M lithium bromide solution at 60 °C for 4 h. This solution was then dialyzed against Milli-Q water for 48 h using a 3500 MWCO cellulose dialysis tubing to remove the lithium bromide. Following dialysis, the solution was centrifuged to separate the silk fibroin (SF), and the clear supernatant was collected to yield a 6 w/v % SF solution, which was stored at 4 °C. For microneedle fabrication, the SF solution was poured into pre-fabricated PDMS molds. These molds were created using laser ablation technology (Blueacre Technology Ltd., Dundalk, Ireland) with specific dimensions tailored for microneedle design. The filled molds were then centrifuged at 805 g for 15 min to ensure the solution settled into the microneedle cavities. The molds were left in a fume hood to allow the SF solution to dry overnight. Once dried, the silk microneedles were carefully peeled from the PDMS molds and immersed in ethanol for 1 h to further induce insolubility. After air-drying at room temperature, the final product was a set of water-insoluble SF microneedles.

## Preparation of TSMNs
Silk microneedles of various sizes were secured in a custom aluminum foil holder, leaving the needles exposed. 158.6 mg (0.450 mmol) of TAPP and 148.2 mg (0.450 mmol) of TFPA were dissolved in 30 mL of a mesitylene/dioxane mixture (v/v, 1/4) in a glass media bottle. 546.6 µL (5.40 mmol) of benzaldehyde and 489.6 µL (5.40 mmol) of aniline were added as competitors. After adding 1.5 mL of 1% (v/v) trifluoroacetic acid in mesitylene/dioxane (v/v, 1/4), the holder was then dipped into this solution with the needles pointing downward. The polymerization was initiated by adding trifluoroacetic acid. After reacting for one day, the microneedles were transferred to a vial filled with ethanol. This vial was subjected to bath sonication for 10–20 min to selectively strip the TAPP-TFPA layer from the substrate while preserving it on the microneedles.

## Plant growth, flavonoid/NBI measurements & drought experiments
Micro Tom tomato seeds were planted in soil and germinated under natural greenhouse conditions. Post-germination, each tomato seedling was relocated to an individual pot and cultivated for one month prior to experimental procedures. Similarly, tobacco seeds were sown and germinated in the same conditions. The sprouted tobacco seedlings were then cleansed of soil and transferred to a hydroponic solution with a pH of 5.6, where they continued to grow for an additional month before experimentation. Plants aged 6–8 weeks were positioned on a growth rack with a 16-h daily lighting schedule at a constant temperature of 25 °C and 65% humidity. They were regularly watered every two days. For drought experiments, watering was ceased, and soil water content was monitored by sampling at a depth of 2 cm below the surface.

Flavonoids and Nitrogen Balance Index (NBI) were chosen as biomarkers for drought stress because of their detectability with portable spectrometers and their significance in plant phenotypes. Flavonoids, vital antioxidants synthesized by plants, help mitigate reactive oxygen species (ROS) produced under abiotic stress conditions. Meanwhile, NBI, indicative of the plant's nitrogen status, typically diminishes during drought conditions to safeguard the photosynthetic apparatus from excessive light damage. Measurement of these two biomarkers were taken using the ForceA Dualex Scientific+™. To reduce measurement errors, readings were taken at multiple positions on the leaves and across different leaves.

## Histology
Smart needles were inserted and then removed from the midribs of tomato or tobacco leaves. The punctured tissues were fixed in 4% formaldehyde for 16 h, followed by a graded ethanol series for dehydration (20%, 40%, 60%, 80%, and then 100% ethanol for 1 h each). Tissues remained in 100% ethanol overnight at 4 °C. The following day, they were processed in a 2:1 ethanol-to-infiltration medium for 2 h, then in a 1:2 ethanol-to-infiltration medium for another 2 h, and finally left in 100% infiltration medium overnight. On the subsequent day, tissues were embedded in the embedding medium and cured in a fume hood for 2 days. The embedded tissues were then secured onto holders and sectioned into 10 µm slices using a microtome. Sections were stained for two minutes with 0.01% toluidine blue, rinsed quickly with water, and imaged with a Zeiss Whitefield microscope.

## pH mapping images
Top-view photographs of COF-silk microneedles were used to generate pH mapping images, utilizing a calibration based on TAPP-TFPA dispersed in various buffer solutions. The images were decomposed into a, b, and L channels according to CIELAB Color Space. These channels were processed and transformed into ΔE maps using ImageJ software. Subsequently, the ΔE maps were converted into pH maps in accordance with the established calibrations.

## Homogeneous nucleation limitation (HNL) strategy for thicker COF film
The LaMer model indicates that early-stage nucleation in a solution generates a high number of homogeneous nuclei, which consume a substantial amount of the COF precursor during the growth phase, thereby limiting the enlargement of heterogeneous nuclei. To address this, a Homogeneous Nucleation Limitation (HNL) strategy can be applied to favor the expansion of heterogeneous nuclei, leading to thicker COF films on substrates. This strategy involves moderating burst nucleation of COFs by adding monofunctional competitors such as aniline and benzaldehyde, which selectively influence homogeneous nucleation without significantly affecting heterogeneous nucleation due to their lower Gibbs free energy. Experimentally, a polyethylene terephthalate (PET) film was submerged in a TAPP-TFPA COF synthetic solution, and polymerization was initiated by the addition of trifluoroacetic acid. The density of homogeneous nuclei was controlled by adjusting the amounts of aniline and benzaldehyde.

Particle coalescence was not considered in the model to simplify the calculation of homogeneous nuclei density, which was determined by measuring the yields and sizes of the COF particles.

## Reporting summary

Further information on research design is available in the Nature Portfolio Reporting Summary linked to this article.

## Data availability

The authors declare that all data supporting the findings of this study are available within the paper and any raw data can be obtained from the corresponding author on request. Source data are provided with this paper.

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

## Acknowledgements

This work was supported by the Singapore-MIT Alliance for Research & Technology (SMART) Centre, National Research Foundation (NRF), Prime Minister's Office, Singapore under its Campus for Research Excellence and Technological Enterprise (CREATE) program. B.M. acknowledges a Paul M. Cook Career Development Professorship and a Singapore Research Professorship. GSV is grateful for support from the National Science Foundation Graduate Research Fellowship Program under Grant No. 2141064. This work was conducted at Disruptive & Sustainable Technology for Agricultural Precision (DiSTAP), an interdisciplinary research group (IRG) from SMART.

## Author contributions

S.W. and M.S. developed the chemical tomography concept. S.W., Y.H., M.S. and B.M. designed the experimental plan and drafted the manuscript. S.W. and Y.H. performed the majority of experiments and data analysis. M.A.C. performed the DFT orbital energy and proton affinity calculations. V.A.R. and J.M.S. did the histology study. Y.C. made the microneedle templates. D.U., S.R., V.A.R. and S.I.L. assisted with plant experiments. P.K.J. and D.T.K. assisted the conversion of 2D and Depth-angel tomographic images of pH mapping. R.C. assisted the microneedle fabrication. G.P.S. assisted the reflection spectrum measurement. G.S.V. assisted the manuscript editing. All authors have revised the manuscript and given their approval of the final version.

## Competing interests

The authors declare no competing interests.
