## [Transparent Peer Review file · Nature Communications]

Chromatic Covalent Organic Frameworks Enabling In-Vivo Chemical Tomography

Corresponding Author: Professor Michael Strano

Version 0:

Reviewer comments:

Reviewer #1

(Remarks to the Author)

The manuscript titled "Chromatic Covalent Organic Frameworks Enabling In-Vivo Chemical Tomography" reported the integration of COFs with silk fibroin microneedles for probing the alkalization of xylem fluids. The design of conical microneedles and the successful film deposition on them provide a solution for 3D mapping of the pH environment in a plant. Although interesting results are presented in this manuscript, the authors should consider the following questions/comments before the next submission:

1. While the authors show delicate designs and systematic results in the work, the fundamental mechanism for the acidochromic properties is not sufficiently discussed. Protonation of the COFs is presumed to be the reason; however, the various nitrogen atoms in the COFs that could potentially be protonated are not thoroughly addressed. I suggest the authors to provide more discussions about the chemical structure's effect on the acidochromic properties, supported by either experimental data or reported references.
2. If the protonation of the COFs is indeed the reason for the acidochromic behavior, the scheme in Figure 1a could be misleading. The current scheme suggests that the introduction of different balls leads to the color variation. It would be more convincing to depict varying amounts of protons for each color region.
3. The authors need to specify the uniqueness of COFs for this application, including advantages over classic litmus paper and network polymers which lack the crystallinity like COFs.
4. Since crystallinity is a crucial property for COFs, the XRD patterns of the COFs with different protonation extents should be provided.
5. The crystallinity of the COFs obtained via the HNL technique should be measured.
6. The authors should confirm the acidochromic reversibility with pH switching, as this is a critical indicator for the application.
7. In the abstract, the authors mentioned "The conical design allows for 3D mapping of the chemical environment at standoff distances from the plant". How could this acidochromic sensor detect the "chemical environment"? Is "pH environment" more suitable here?
8. There are numerous spelling errors in the manuscript. For examples, page 4, "Rational Design of the pH-sensitive Imide COF Series", "imide" should be "imine"; Page 7, in the figure caption of Figure 2, "I Experimental and predicated pKa of different COFs", "I" should be "(E)"; Page 12, "At 450 um, the pH shifted from 5.5 to 6.5 within 10 minutes. In contrast, at 250 um, the pH stayed constant after 30 minutes, underscoring the microneedle's ability to access specific tissue layers within the living leaf.", "um" should be "µm"; Page 15, "Photographic evidence (D) also corroborates the water stresslstate", what is "stresslstate"?; The caption of Figure S14, "SEM images of TAPP-TFPA film growing on BET substrates", is "BET" substrate correct? The authors should carefully check/correct the manuscript before the next submission.

Reviewer #2

(Remarks to the Author)

The authors present a nice approach for combining color changing coatings with transparent microneedles to monitor depth-dependent changes in chemistry (in this case pH) within a leaf vein of an intact plant experiencing drought stress. Those aspects of the research are clear and convincing. It is also very important and very useful to be able to separate what is happening at the epidermis versus deeper in leaf tissues as was also well demonstrated. However, the utility of the method had not been adequately demonstrated for scientific value to basic plant research or to agricultural applications as described

below.

My concerns with this work are tied to the descriptions of what is being monitored and the utility of the method as numbered below.

1) The methods presented would only measure fluids in and around vascular tissues as a whole (images clearly contact both xylem and phloem in the vascular bundle). There is no evidence for xylem-specific measurements and any mention of xylem sap or other xylem specific detection should be removed. Xylem will deliver solutions to the leaves and then that water moves to other cells in the vascular bundles and out to the rest of the plant. The sensors would detect the changes wherever they occur. Furthermore, if the needles were in xylem, those cells would likely embolize and stop transporting water.

2) Figure 5 shows that this method doesn't detect drought until day 4, when the soil had lost nearly two-thirds of the water and the plants were visibly droughted. Thus, there isn't any clear value to using this method over just visually observing plant wilting. There may be other chemistries that could detect changes occurring over hours or in the first day or two that would have great scientific and practical value, but the pH monitoring doesn't provide those benefits, partially because a greater pH resolution would be needed. Also, it is not clear why figure 5a-d show 9 days of data but 5g-i only show 6 days. Either add the remaining 3 days to g-i or remove them from a-d.

3) Although I wouldn't expect significant damage or wound responses over the examined 4 days, I am not convinced that the method used is valid. It is not clear what area was analyzed in Fig 4b. How much of the NBI sensor's field of view was covered by the microneedle location? Could the method even detect damage from the needle if it occurred? Also, why stop at 4 days when the later experiments with drought go to 6 or 9 days? Why not test over weeks or months by just leaving needles embedded. That is what would be needed in many applications.

4) It is not clear how the needles remained attached and in close contact with the leaf material. Over time, these needles should loosen and fall out on their own.

5) It would be helpful to have a clearer analysis of the impact of viewing angle on the analyses. How carefully does one have to align the leaf/needles with the camera to get the proper data?

6) Need scale bars in Fig 1 and something to help show where the needles are in the larger image of tomato leaves.

Version 1:

Reviewer comments:

Reviewer #1

(Remarks to the Author)

The manuscript has been improved. My recommendation is Accept.

Reviewer #3

(Remarks to the Author)

[Note from the Editor: Reviewer #3 and Reviewer #4 were asked to assess the responses given to the original Reviewer #2. Two new reviewers were invited to ensure the expertise of reviewers covers all the technical aspects in this work.]

The authors have completed the corresponding work based on the comments of Reviewer #2, but I think the authors have not fully addressed the issues raised by the Reviewer #2. Based on feedback from Reviewer #2 and the authors, and considering the research content of the manuscript, we present the following questions and comments:

(1) Regarding the first question raised by Reviewer #2, the authors did not fully explain the effect of microneedles on xylem embolism. The cross-sectional view in Figure 5E demonstrates that a small part of the xylem is affected but does not quantify the extent of the impact of microneedles on water transport. Additionally, the authors' previous work (Adv. Sci. 2020, 7, 1903551) suggested that insoluble matter in plant sap may hinder sap flow in the xylem and phloem through vascular obstruction. They proposed that hydrophilic silk fibroin-derived peptides (Cs) were extracted to increase the hydrophilic content of the silk end material and reduce the size of protein biodegradation byproducts. However, since the microneedles in this article do not involve Cs, this previous work cannot strongly support the relevant viewpoints.

(2) Regarding the second question raised by Reviewer #2, the authors did not clarify the necessity of pH monitoring to indicate drought stress. Drought stress can reduce chlorophyll levels, thereby decreasing the amount of photons absorbed by leaves. It may also lead to excessive accumulation of excitation energy, resulting in the formation of reactive oxygen species (ROS). Why not monitor chlorophyll or ROS levels instead?

(3) Regarding the third question raised by Reviewer #2, the authors cited previous work to demonstrate that the microneedles used did not cause significant damage to plants at that time. However, this does not imply that all types of microneedles are harmless to plants, as their composition and structure can vary. Moreover, relying solely on the NBI level is inadequate to establish that the microneedles used in this manuscript do not cause significant harm to plants. Could the authors conduct additional relevant experiments to demonstrate this?

(4) Regarding the fourth question raised by Reviewer #2, the authors mentioned that for studies on a longer time scale, new microneedles were injected into the plants daily. How can the influence of individual differences in microneedles on the test results be mitigated? What are the stability and repeatability of microneedle detection?

(5) Regarding the fifth question raised by Reviewer #2, the authors used a mobile phone to photograph the microneedles. However, direct handheld photography may introduce human error. How can the deviation in viewing angle be kept within approximately 9°?

(6) The microneedle was used for tomographic chemical imaging of tomato plants under drought stress. Relevant references indicated that the pH value of xylem sap changes significantly from the root to the canopy at the onset of drought stress. Why did you choose tomato leaves as the detection site instead of roots or stems?

(7) The authors conducted a detailed analysis of the COF structure and its thickness control but did not thoroughly evaluate the pH detection capability of microneedle with COF and microneedle integration. Instead, they applied it directly to pH detection in plants, which may not sufficiently support the in vivo detection results.

Reviewer #4

(Remarks to the Author)

[Note from the Editor: Reviewer #3 and Reviewer #4 were asked to assess the responses given to the original Reviewer #2. Two new reviewers were invited to ensure the expertise of reviewers covers all the technical aspects in this work.]

The manuscript by Wang et al., titled "Chromatic Covalent Organic Frameworks Enabling In-Vivo Chemical Tomography," reports the synthesis of COF coatings for assembling color-changing microneedles. When inserted into plant leaf tissue, these microneedles allowed for 3D in vivo monitoring of pH modulations associated with drought stress. The innovative results are presented in a clear and comprehensive manner.

As requested, I will focus primarily on evaluating the authors' responses to Reviewer #2. The authors have reasonably addressed most questions, overall improving the manuscript. However, a few aspects should be clarified before publication:

1. The authors' claim to map the chemical environment should be supported by a detailed discussion on how the present COF network could be modified to sense other analytes beyond proton concentrations. This will strengthen the applicability and outline a general design concept.

2. Are other analytes expected to accumulate or change during drought stress that could influence the colorimetric readout? Was the selectivity of the COF microneedles tested, for example, against different phytohormones or varying ion concentrations?

3. While the presented approach is scientifically highly interesting for monitoring pH changes in a 3D manner with high spatio-temporal resolution, the proposed application as an agricultural diagnostic tool seems exaggerated or at least needs clearer discussion. The authors state: "These findings underscore that water stress indicators become discernible 48 hours prior to detection via prevailing methodologies in plant science and agronomy." To which conventional methods do the authors refer? How do the colorimetric sensors compare to fast drought stress analyses, such as the assessment of leaf water potential?

Version 2:

Reviewer comments:

Reviewer #3

(Remarks to the Author)

The manuscript has been improved. I recommend accepting it for publication.

Reviewer #4

(Remarks to the Author)

I recommend the publication of the manuscript in its current form.

Reviewer #1 (Remarks to the Author):

The manuscript titled “Chromatic Covalent Organic Frameworks Enabling In-Vivo Chemical Tomography” reported the integration of COFs with silk fibroin microneedles for probing the alkalization of xylem fluids. The design of conical microneedles and the successful film deposition on them provide a solution for 3D mapping of the pH environment in a plant. Although interesting results are presented in this manuscript, the authors should consider the following questions/comments before the next submission:

R: We thank the reviewer for the constructive comments. We addressed all of them using additional experimental data and significant revisions to the manuscript text.

1. While the authors show delicate designs and systematic results in the work, the fundamental mechanism for the acidochromic properties is not sufficiently discussed. Protonation of the COFs is presumed to be the reason; however, the various nitrogen atoms in the COFs that could potentially be protonated are not thoroughly addressed. I suggest the authors to provide more discussions about the chemical structure’s effect on the acidochromic properties, supported by either experimental data or reported references.

R: With regards to fundamental mechanism for the COF acidochromic properties, we have found that the imine nitrogen is the first protonation point when present together with pyridine or triphenylamine nitrogen as reported in *Journal of the American Chemical Society*, 2009, 131(43): 15815-15824 and *Angewandte Chemie International Edition*, 2021, 60(36): 19797-19803. In response to the reviewer’s suggestion, we have clarified the above and referenced the two publications in the main text (Page 5, Line 81-82).

2. If the protonation of the COFs is indeed the reason for the acidochromic behavior, the scheme in Figure 1a could be misleading. The current scheme suggests that the introduction of different balls leads to the color variation. It would be more convincing to depict varying amounts of protons for each color region.

R: We thank the reviewer for this constructive comment. In response to the reviewer’s comment, we modified the illustration to avoid misleading the readers. Specifically, we use relative concentrations of proton and hydroxide to show different pH. Please see the updated Figure 1a.

3. The authors need to specify the uniqueness of COFs for this application, including advantages over classic litmus paper and network polymers which lack the crystallinity like COFs.

R: In response to the reviewer's comment, we have expanded the discussion about uniqueness of COFs for this application in the revised manuscript (Page 4 – 5, Line 74-77). The abundant nanopores of crystalline COFs can dramatically decrease the diffusion resistance of guest molecule, thus enable the ultrafast chromic response. As an insoluble 2D polymer, COFs will not leak small molecular dyes that could potentially cause contamination in plant tissues.

4. Since crystallinity is a crucial property for COFs, the XRD patterns of the COFs with different protonation extents should be provided.

R: In response to the reviewer's concern on this point, we have now performed additional XRD experiments to characterize the COF crystallinity at different pHs (Figure S7). Specifically, we dispersed the TAPP-TFPA COF powders in different pH buffer aqueous solutions for 1 day, and characterized their crystallinity using XRD. Here, we show that the COF crystallinity is maintained when it is subjected to different protonation extents. The manuscript has been amended accordingly at Page 5, Line 86-87.

Figure S7. XRD spectrum of TAPP-TFPA powders dispersed in different buffer aqueous solutions for 1 day.

5. The crystallinity of the COFs obtained via the HNL technique should be measured.

R: The XRD spectrum of COFs in Figure S2-5 are synthesized with HNL technique with different competitor (C) amount, please see Table 1 in Methods section. To demonstrate how the HNL technique influences COF crystallinity, we performed additional XRD experiments to test the crystallinity of TAPP-TFPA with different C amount, as shown below in Figure S17. With increase in C amount, we show that crystallinity of TAPP-TFPA COF has obviously increased. Hence, HNL technique can not only increase COF film thickness, but also increase its crystallinity. For the COF-microneedle fabrication in this manuscript, 12 C amount is used to achieve high crystallinity. The manuscript main text is revised accordingly at Page 8, Line 154-155.

Figure S7. XRD spectrum of TAPP-TFPA synthesis with different competitor amounts (HNL technique).

6. The authors should confirm the acidochromic reversibility with pH switching, as this is a critical indicator for the application.

R: In response to the reviewer’s comment, we conducted an additional experiment to test the acidochromism of the TAPP-TFPA COF powder. We switched the buffers between pH 4.0 (acid) and pH 9.0 (base) by centrifuging the dispersion. As presented in Figure S9, after 10 repeated cycles of acid-base buffer switching, the chromic performance remains stable. We have revised the manuscript accordingly at Page 7, Line 123-124.

Figure S9. Color of TAPP-TFPA COF powder aqueous dispersion during 10 times buffer switching.

7. In the abstract, the authors mentioned “The conical design allows for 3D mapping of the chemical environment at standoff distances from the plant”. How could this acidochromic sensor could detect the “chemical environment”? Is “pH environment” more suitable here?

R: We agree with the reviewer’s comment and have amended the abstract to state that “the conical design allows for 3D mapping of the chemical

environment (such as pH) at standoff distances from the plant". In this manuscript, proton concentration changes induced chromic responses in the COF microneedles. In theory, COF chemistries can also be leveraged to design chromic sensors for other chemical changes in the plant environment, besides proton concentration changes.

8. There are numerous spelling errors in the manuscript. For examples, page 4, "Rational Design of the pH-sensitive Imide COF Series", "imide" should be "imine"; Page 7, in the figure caption of Figure 2, "I Experimental and predicated pKa of different COFs", "I" should be "(E)"; Page 12, "At 450 um, the pH shifted from 5.5 to 6.5 within 10 minutes. In contrast, at 250 um, the pH stayed constant after 30 minutes, underscoring the microneedle's ability to access specific tissue layers within the living leaf.", "um" should be "µm"; Page 15, "Photographic evidence (D) also corroborates the water stresslstate", what is "stresslstate"?; The caption of Figure S14, "SEM images of TAPP-TFPA film growing on BET substrates", is "BET" substrate correct? The authors should carefully check/correct the manuscript before the next submission.

R: We thank the reviewer for pointing out these typo errors. We have corrected all of them, and have carefully proof read the manuscript and SI files.

Reviewer #2 (Remarks to the Author):

The authors present a nice approach for combining color changing coatings with transparent microneedles to monitor depth-dependent changes in chemistry (in this case pH) within a leaf vein of an intact plant experiencing drought stress. Those aspects of the research are clear and convincing. It is also very important and very useful to be able to separate what it happening at the epidermis versus deeper in leaf tissues as was also well demonstrated. However, the utility of the method had not been adequately demonstrated for scientific value to basic plant research or to agricultural applications as described below.

R: We thank the reviewer for all the constructive comments, which dramatically improved our manuscript. As the reviewer mentioned, conical sensing technology for chemical spatial mapping is the main advantage of our work. As demonstration of this capability, we monitored the pH change in plant vascular system. With additional experiments and significant revision to the manuscript, we have addressed the concerns raised by the reviewer below.

My concerns with this work are tied to the descriptions of what is being monitored and the utility of the method as numbered below.

1)The methods presented would only measure fluids in and around vascular tissues as a whole (images clearly contact both xylem and phloem in the

vascular bundle). There is no evidence for xylem-specific measurements and any mention of xylem sap or other xylem specific detection should be removed. Xylem will deliver solutions to the leaves and then that water moves to other cells in the vascular bundles and out to the rest of the plant. The sensors would detect the changes wherever they occur. Furthermore, if the needles were in xylem, those cells would likely embolize and stop transporting water.

R: We thank the reviewer for this suggestion to remove all suggestions of xylem specific detection or measurement. In place of that, we have referred to “vascular tissues” as a whole in the revised manuscript. The xylem system transports water and soluble mineral nutrients from the roots throughout the plant. The microneedle tip is cone-shaped and upon injection, only a small part of xylem will be affected. This is confirmed by our cross-section images in Figure 5E. As such, xylem cells are unlikely to embolize and stop transporting water. In Figure 4 D-F, we have shown that the pH of vascular tissues of tobacco leaf changes when plants are switched between acidic and basic pH buffer solutions. This demonstrates that the needles do not obstruct water transport in the plant vascular system.

Furthermore, our lab had previously reported that microneedles can deploy payloads that were translocated along the vascular tissue (Adv. Sci. 2020, 7, 1903551), as shown in Reviewer-only Figure below. This further illustrates that the microneedle injection does not block water transport in plants.

Figure R1. Image assembly of fluorescent images showing 5(6)-carboxyfluorescein diacetate delivered to and transported in xylem, from roots to canopy, 1, 3, and 5 min postinjection. (Adv. Sci. 2020, 7, 1903551)

2) Figure 5 shows that this method doesn't detect drought until day 4, when the soil had lost nearly two-thirds of the water and the plants were visibly droughted. Thus, there isn't any clear value to using this method over just visually observing plant wilting. There may be other chemistries that could detect

changes occurring over hours or in the first day or two that would have great scientific and practical value, but the pH monitoring doesn't provide those benefits, partially because a greater pH resolution would be needed. Also, it is not clear why figure 5a-d show 9 days of data but 5g-i only show 6 days. Either add the remaining 3 days to g-i or remove them from a-d.

R: We thank the reviewer for the comment. In drought stress experiment, at Day 1, we have saturated the soil with water and subsequently stopped irrigation. Water loss of the soil was calculated and shown in Figure 5a. For most varieties of tomatoes, drought stress typically occurs when the soil moisture content drops below a certain threshold, usually around 50% of the soil's water holding capacity (Frontiers in Plant Science, 2023, 14: 1211210). In Day 3, the soil moisture content was measured to be 2.5 g water/g soil, corresponding to 35% water loss. Hence, at Day 3, the soil moisture content remains above the threshold that is indicative of drought stress. In Day 4, water loss reaches 57%, indicating that the plant has begun experiencing drought stress. On the same day (Day 1 of drought stress), our pH sensor detects commencement of acidity reduction. In contrast, visible plant yellowing resulting from drought stress can only be observed at Day 6 (Day 3 of drought stress). For clarity, we have modified the manuscript Page 16, Line 324-326. For consistency, as suggested by the reviewer, the data of the days 7-9 have been removed from Figure 5a-d, and the plant images were added in Figure 5d.

3)Although I wouldn't expect significant damage or wound responses over the examined 4 days, I am not convinced that the method used is valid. It is not clear what area was analyzed in Fig 4b. How much of the NBI sensor's field of view was covered by the microneedle location? Could the method even detect damage from the needle if it occurred? Also, why stop at 4 days when the later experiments with drought go to 6 or 9 days? Why not test over weeks or months by just leaving needles embedded. That is what would be needed in many applications.

R: To clarify, the NBI sensor's field of view was not covered by the microneedle location. In Figure S28 below, we illustrate that the microneedle was injected at the base of the mid rib (red circle), while the NBI sensor's field of view was at the five different positions of the leaf as indicated by the blue circles. If the microneedle injection at the mid-rib had caused significant wounding, the entire leaf health will have been affected. Following the reviewer's suggestion, we have extended the NBI measurement to 15 days. The data in Figure 4C, shows that there is no obvious difference in NBI between injected leaves and control leaves. The manuscript text is amended accordingly at Page 11, Line 223.

Our group previous reported work demonstrated the wounding effects of

microneedles in a wild-type plant of high agricultural importance (Advanced Materials, 2023, 35(2): 2205794). we performed time-course Quantitative Reverse Transcription-Polymerase Chain Reaction (RT-qPCR) experiments using wild-type *L. sativa* var. Parris Island. Lettuce plants were injected with GA3-loaded microneedles (GA_INJ) or microneedles without GA (Mock), and samples were harvested for RT-qPCR analyses 1, 3, or 24 h following microneedle injection. The wounding response marker genes used are closest lettuce homologs of the Arabidopsis wounding DEGs found in our *ft-10* transcriptome. Gene expression data were normalized to Control (non-treated) plants collected at their respective time points (1, 3, or 24 h). In both Mock and GA_INJ groups, across all genes tested, we observed an increase in gene expression 1 h or 3 h after microneedle injection (Figure R2 below). Expression levels of wounding markers returned to control levels 24 h after microneedle injection. Wounding effects induced by silk-based microneedles were minimized within 24 h, suggesting that the use of microneedles is minimally invasive and highly effective.

Our group recent work (*unpublished data for reviewer only*) has studied the impact of different types of microneedles on tomato plant. 4-5 weeks-old tomato plants were injected with hollow, porous, and solid microneedles (HMN, PMN, and SMN, respectively). As indicated by trypan blue staining for dead cells, local cell death and scar formation at injection sites were observed (Figure R3). The impact of all microneedles injections on the whole plant is negligible 14 days post-injection (Figure R4). Measurements of plant height, chlorophyll content (using a SPAD meter), stomatal conductance (using a LICOR LI-600), and quantum efficiency of photosystem II (using a LICOR LI-600) post-injection were used to evaluate the systemic impact of microneedle injection on plant health (Figure R4). These phenotypic differences between injected and untreated plants were not statistically significant ($p > 0.05$).

Figure S28. Microneedle injection point and NBI detection points for the study of healthy influence of microneedle on tobacco.

Figure 4C. Nitrogen balance index (NBI) measured in tobacco leaves injected by a TSMN700 to monitor its health while carrying the microneedle up to day 15.

Figure R2. RT-qPCR quantification of gene responses following microneedle injection in *Lactuca sativa*. a-h. Relative expression levels of wounding response genes in

Control, Mock and GA3-injected *L. sativa* (lettuce) plants after 1, 3, or 24 h after microneedle injection. Gene expression changes are calculated and represented as mean relative expression to Control (non-injected) plants at each timepoint collected. Data shown are mean \pm s.d. for $n = 3$ biological replicates within each group. One-way ANOVA was performed, followed by the Tukey test (a-b, d-h). One-way ANOVA assuming unequal variances was used, followed by the Games-Howell test if homogeneity of variances or equal group size was violated (c). * $p < 0.05$, ** $p < 0.01$, *** $p < 0.001$, **** $p < 0.0001$. (Advanced Materials, 2023, 35(2): 2205794)

Figure R3. Phenotypic local wounding post-injection. a, Solid, hollow, and porous microneedles injected on tomato petioles. b, Tomato petioles in a after removing the microneedles. Red arrows indicate the injection sites. Scale bar 5 mm. c-f, Tomato petioles post-injection on day 0 (c), day 1 (d), day 7 (e), and day 14 (f) by different types of microneedles, stained by trypan blue. The blue color indicates dead cells.

Scale bar 1 mm. (*unpublished data for reviewer only*)

Figure R4. Phenotypic systemic wounding post-injection. a. Representative images of plants after microneedle injection. b. Plant height measurements 5 days after microneedle injection ($n = 8$). c-e. Phenotypic observation of plant health following microneedle injection ($n = 8$). Traits measured include relative chlorophyll content (c), stomatal conductance (d), and quantum efficiency of photosystem II (e). The violin plots show the distribution of data points, while the boxplot shows minimum and maximum non-outlier observations (whiskers), lower quartile (Q1), median (Q2), and upper quartile (Q3). One-way ANOVA assuming equal variances was performed, followed by the Tukey test. * $p < 0.05$, ** $p < 0.01$. (*unpublished data for reviewer only*)

4) It is not clear how the needles remained attached and in close contact with the leaf material. Over time, these needles should loosen and fall out on their own.

R: We agree with the reviewer that the microneedle will loosen and lose its close contact with the leaf material after about one day. Hence, for longer time-scale studies such as the detection of drought stress over days in Figure 5, we have injected new microneedles every day into the plants. For shorter time-scale studies that are completed within a few hours, such as those in Figure 4, we have collected the data from the same microneedle. For clarity, we modified the manuscript at Page 16, Line 301-303 accordingly.

5) It would be helpful to have a clearer analysis of the impact of viewing angle on the analyses. How carefully does one have to align the leaf/needles with the camera to get the proper data?

R: In response to the reviewer's comment, we have added a simulation to demonstrate the impact of viewing angle on the analyses (Figure S24). Given the 3D geometry of microneedle, top views of the microneedle with different viewing angle deviation are simulated and grouped into 3 different categories: separate, tangent, and intersecting. At viewing angle deviation of $\sim 10^\circ$, the inner circle that reaches deeper into the plant vascular system is now tangent to the outer circle, which means that one side of the information is lost due to the overlapping. Hence, for accurate monitoring of pH spatial distribution in plant vascular tissues, the viewing angle deviation need to be kept within $\sim 9^\circ$. The manuscript main text is revised accordingly at Page 10, Line 203-205.

Figure S24. Simulated top views of the microneedle with different viewing angle deviation.

6)Need scale bars in Fig 1 and something to help show where the needles are in the larger image of tomato leaves.

R: We thank the reviewer for this comment. We have added scale bar and marked the position of microneedles in Figure 1 with a red circle.

Reviewer #1 (Remarks to the Author):

The manuscript has been improved. My recommendation is Accept.

R: We thank the reviewer for the acceptance recommendation.

Reviewer #3 (Remarks to the Author):

[Note from the Editor: Reviewer #3 and Reviewer #4 were asked to assess the responses given to the original Reviewer #2. Two new reviewers were invited to ensure the expertise of reviewers covers all the technical aspects in this work.]

The authors have completed the corresponding work based on the comments of Reviewer #2, but I think the authors have not fully addressed the issues raised by the Reviewer #2. Based on feedback from Reviewer #2 and the authors, and considering the research content of the manuscript, we present the following questions and comments:

R: Thanks for the constructive comments. We address all of them to eliminate the concerns of the reviewer. Please see the details below.

(1) Regarding the first question raised by Reviewer #2, the authors did not fully explain the effect of microneedles on xylem embolism. The cross-sectional view in Figure 5E demonstrates that a small part of the xylem is affected but does not quantify the extent of the impact of microneedles on water transport. Additionally, the authors' previous work (Adv. Sci. 2020, 7, 1903551) suggested that insoluble matter in plant sap may hinder sap flow in the xylem and phloem through vascular obstruction. They proposed that hydrophilic silk fibroin-derived peptides (Cs) were extracted to increase the hydrophilic content of the silk end material and reduce the size of protein biodegradation byproducts. However, since the microneedles in this article do not involve Cs, this previous work cannot strongly support the relevant viewpoints.

R: As suggested by the reviewer, we tested the water transport in the plant leaf injected with microneedle (Figure S31). Tobacco leaf with and without microneedle injection was cut off after submerging the petioles into rhodamine B (100 μ M) solution for 5 minutes. The rhodamine B solution was absorbed by the tobacco leaves. The petiole of leaves was cut at 5 mm above and below the injection site and the bottom and up cross-section was observed by microscope (left and middle photos in Figure S31 C and D). The vascular bundles of the petiole, whether injected with microneedle or not, were stained with rhodamine B. We also show the rhodamine B transported and remained in the vascular bundles of the midrib at 5-10 mm above the injection point (right photos in Figure S31 C and D). This demonstrates that the needles do not obstruct water transport in the plant vascular system. We add the figure to SI as Figure S31 and discussion at Page 11 Line 230-232 in the main text.

Figure S31. Digital photos showing the rhodamine B transporting in tobacco leaves. Tobacco leaf with and without microneedle injection was cut off and its petiole was submerged into rhodamine B (100 μ m) solution for 5 minutes (A). The rhodamine B solution was absorbed by the tobacco leaves (B). The petiole of leaves with or without microneedle injection was cut at 5 mm above and below the injection site and the cross-section was observed by microscope (left and middle figures in C and D). Rhodamine B transported and remained in the vascular bundles of the midrib at 5-10 mm above the injection point (right photos in C and D). Vascular bundles of the midrib were exposed by cutting. The vascular bundles of the petiole, whether injected with microneedle (C) or not (D), were stained with rhodamine B.

(2) Regarding the second question raised by Reviewer #2, the authors did not clarify the necessity of pH monitoring to indicate drought stress. Drought stress can reduce chlorophyll levels, thereby decreasing the amount of photons absorbed by leaves. It may also lead to excessive accumulation of excitation energy, resulting in the formation of reactive oxygen species (ROS). Why not monitor chlorophyll or ROS levels instead?

R: We thank the reviewer for the suggestion of chlorophyll or ROS detection. The major contribution of this interdisciplinary work is developing a conical sensing technology for chemical spatial mapping in intact plant, not just targeting at drought stress detection. It's a new formation or platform for in planta chemicals detection, and pH mapping/monitoring is employed to demonstrate and prove the concept. Up to now, it's

still challenging to integrate selective and sensitive detection of ROS or chlorophyll into covalent organic frameworks (COFs). However, our lab developed some other nanomaterial, e.g., carbon nanotube based nanosensors for ROS (Nature plants, 2020, 6(4): 404-415.) or phytohormones such as salicylic acid (Nature Communications, 2024, 15(1): 2943) and Auxins (ACS sensors, 2021, 6(8): 3032-3046.). In the future, based on the conical sensing technology developed in this paper, we are going to integrate our carbon nanotube sensors for more abundant physiological chemical mapping, targeting at biotic and abiotic stress detection of plants.

(3) Regarding the third question raised by Reviewer #2, the authors cited previous work to demonstrate that the microneedles used did not cause significant damage to plants at that time. However, this does not imply that all types of microneedles are harmless to plants, as their composition and structure can vary. Moreover, relying solely on the NBI level is inadequate to establish that the microneedles used in this manuscript do not cause significant harm to plants. Could the authors conduct additional relevant experiments to demonstrate this?

R: As suggested by Reviewer, we evaluate the impact of microneedle in plant leaf, by chlorophyll fluorescence imaging analysis. The tobacco leaves and after the injection of microneedle for 1h and 24 h, were cut for the measurements. Leaves without any injection are as control. The leaves without the microneedle were used as control. We can see high F_v/F_m ratios (> 0.8) of control, 1h and 24h leaf, indicating that the leaves are healthy with efficient photosynthesis. The NPQ values of microneedle leaves remained at the same level as those of control leaves (below 3), that demonstrate the healthy state of leaves with the ability to dissipate excess light energy as heat. Φ_{PSII} values of all leaves indicate that a significant portion of absorbed light energy is being used effectively for photochemistry. These results show the plant leaves after the microneedle injection are healthy. We put the data in the SI as Figure S30 and discussion at Page 11 Line 228-230 in the main text.

B

Runs	F_v/F_m			Φ_{PSII}			NPQ		
	Control	1 h	24 h	Control	1 h	24 h	Control	1 h	24 h
1	0.82	0.82	0.82	0.39	0.49	0.49	1.36	0.90	0.80
2	0.81	0.81	0.82	0.46	0.48	0.51	0.77	0.67	0.76
3	0.82	0.81	0.83	0.55	0.47	0.53	0.63	0.90	0.74
avg.	0.82	0.81	0.82	0.47	0.48	0.51	0.92	0.82	0.77

Figure S30 (A) Representative images of the digital and symptomatology photos of control leaf (without injection) and experimental leaves after injection of microneedle after 1 h and 24 h. (B) PSII efficiency (F_v/F_m and Φ_{PSII}) and non-photochemical quenching (NPQ) values are summarized in the table.

(4) Regarding the fourth question raised by Reviewer #2, the authors mentioned that for studies on a longer time scale, new microneedles were injected into the plants daily. How can the influence of individual differences in microneedles on the test results be mitigated? What are the stability and repeatability of microneedle detection?

R: For the drought stress detection in the last figure, for each day, we do the injection by four replicates using four different new microneedles. The data is summarized in the Figure 5G in the main text. As you can see, the pH detection by different microneedles is quite stable and repeatable. The pH increasing trend is clear and the statistical significance analysis based on those replicates shows the pH detection by the microneedles is reliable.

Figure 5G. pH of vascular saps measured by inner circle color difference against outer circle for TSNM700 injected into the midrib of tomato leaves at different time points, after day 1. Statistical significance: ns is $p > 0.05$, *** is $p < 0.001$.

(5) Regarding the fifth question raised by Reviewer #2, the authors used a mobile phone to photograph the microneedles. However, direct handheld photography may introduce human error. How can the deviation in viewing angle be kept within approximately 9° ?

R: Thanks for the comment and we add more simulated data and experimental data to clarify this. As shown in the Figure S24 B, From the perfect vertical view, the edge of the microneedle substrate cannot be observed. With the viewing angle deviation increases, more and more edge can be seen. In the experiment, we utilized this mechanism to get vertical view. As shown in the Figures24 C, when we took the photos, we tried our best to minimum substrate edge showing up, so that we can get the viewing angle deviation small enough. We add data as Figure S25 and the discussion at Page 10 Line 207-209 in the main text.

Figure S25 (A) Simulated top views of the microneedle with different viewing angle deviation. At viewing angle deviation of $\sim 10^\circ$, the circles tangent, which means one side of the information is missing due to the overlapping. So for the microneedle we used, the viewing angle deviation need to be kept within $\sim 9^\circ$. (B) Simulated top views of the microneedle substrate edge with different viewing angle deviation edge. From the perfect vertical view, the edge of the microneedle substrate can not be observed. With the viewing angle deviation increase, more and more edge can be seen. (C) Experimental microscope photo of TSMN700 injecting into tobacco leaf. In experiment, we try our best to get the top view photos with minimum substrate edge showing up.

(6) The microneedle was used for tomographic chemical imaging of tomato plants under drought stress. Relevant references indicated that the pH value of xylem sap changes significantly from the root to the canopy at the onset of drought stress. Why did you choose tomato leaves as the detection site instead of roots or stems?

R: To detected the pH change of xylem sap change, we need to inject our microneedle to vascular bundle region. And the vascular bundle is centralized in the middle, however, decentralized in the stem, as shown below. It's much easier for us to reach vascular bundle by injecting at leaves. Furthermore, from the view of practical application, it's not convenient to get the roots out of the soil for each detection. Overall, we finally chose leaves as the detection site.

Figure for review only. Vascular bundle distribution in midrib and stem of tomato. (Brazilian Archives of Biology and Technology, 2020, 63: e20180670.)

(7) The authors conducted a detailed analysis of the COF structure and its thickness control but did not thoroughly evaluate the pH detection capability of microneedle with COF and microneedle integration. Instead, they applied it directly to pH detection in plants, which may not sufficiently support the in vivo detection results.

R: As the reviewer said, the pH detection capability does show slight difference with COF films or powder. We did the thoroughly evaluation and the data were summarized in Figure S25 and Figure S26 in the supplementary. The COF-microneedle shows a pKa of 6.0 (Figure S26), slightly lower than 6.6 of powder and larger than 5.6 of film. And it shows a fast chromic response time of 8 seconds (Figure S27 and SI video). The discussions are at Page 10, Line 209-210.

Figure S26 Top view optical microscope images (A) and ΔE (B) of TSMN700 upon exposure to buffers with pH values varying from 4.0 to 7.5. ΔE is calculated based on TSMN700 color at pH = 7.5 buffer.

Figure S27 Side views optical microscope images (A) and ΔE (B) of TSMN700 as a function of time upon insertion into a transparent acidic gel (pH 5). TSMN700 is directly drying from ethanol without any treatment. ΔE is calculated based on initial TSMN700.

Reviewer #4 (Remarks to the Author):

[Note from the Editor: Reviewer #3 and Reviewer #4 were asked to assess the responses given to the original Reviewer #2. Two new reviewers were invited to ensure the expertise of reviewers covers all the technical aspects in this work.]

The manuscript by Wang et al., titled “Chromatic Covalent Organic Frameworks Enabling In-Vivo Chemical Tomography,” reports the synthesis of COF coatings for assembling color-changing microneedles. When inserted into plant leaf tissue, these microneedles allowed for 3D in vivo monitoring of pH modulations associated with drought stress. The innovative results are presented in a clear and comprehensive manner.

As requested, I will focus primarily on evaluating the authors' responses to Reviewer #2. The authors have reasonably addressed most questions, overall improving the manuscript. However, a few aspects should be clarified before publication:

R: We thank the reviewer for the recognition of our work and all the constructive comments. And we address all of them, please see the details below:

1. The authors' claim to map the chemical environment should be supported by a detailed discussion on how the present COF network could be modified to sense other analytes beyond proton concentrations. This will strengthen the applicability and outline a general design concept.

R: Thanks for the suggestion. We add a discussion about the reported biosensing

performances of COF in the main text following the reviewer's comment, which is "COFs are emerging candidates for biosensor materials. Recent advances have shown their succussed in biosensing with targets of DNA,¹¹ antibiotics,¹⁵ as well as biomarkers,¹⁶ showing the promise for selective detection of physiological molecules of plant." Please find it at Page 4, Line 72-75 in the main text.

2. Are other analytes expected to accumulate or change during drought stress that could influence the colorimetric readout? Was the selectivity of the COF microneedles tested, for example, against different phytohormones or varying ion concentrations?

R: Following the suggestion of the reviewer, we dispersed the TAPP-TFPA COF powder in sodium acetate buffer solutions (pH=5.2) with different concentrations from 0.1 M to 2 M. We also add two of the phytohormones, i.e., indole-3-acetic acid (IAA) and salicylic acid (SA) with a high concentration of 1 mg/mL to the buffer solution. As presented below, no color difference is observed between them. Based on the data, we can draw the conclusion that different phytohormones or varying ion concentrations have no influence on the chromatism of the COF we used. We add the discussion at Page 8 Line 128-129 in the main text and the figure as Figure S10 in the supplementary.

Figure Photos of TAPP-TFPA COF powders dispersed in sodium acetate buffer solutions (pH=5.2) with different concentrations from 0.1 M to 2 M, as well as the present of indole-3-acetic acid (IAA) and salicylic acid (SA) with a concentration of 1 mg/mL.

3. While the presented approach is scientifically highly interesting for monitoring pH changes in a 3D manner with high spatio-temporal resolution, the proposed application as an agricultural diagnostic tool seems exaggerated or at least needs clearer discussion. The authors state: "These findings underscore that water stress indicators become discernible 48 hours prior to detection via prevailing methodologies in plant science and agronomy." To which conventional methods do the authors refer? How do the colorimetric sensors compare to fast drought stress analyses, such as the assessment of leaf water potential?

R: The COF-microneedle platform is optical-based sensing, which possess apparent advantages of easy operation, fast data collecting and nondestructive. Furthermore, it's almost device free, only needs a smartphone with a cheap and portable lens. So here,

the conventional methods we use as control are also portable optical-based, i.e. portable spectrometers (ForceA Dualex Scientific+™, for the detection of flavonoid absorption and nitrogen balance index) and photographing (for the detection of yellowing of leaves), which is widely used in the agriculture science. We modified the discussion in the main text as “These findings underscore that water stress indicators become discernible 48 hours prior to detection via prevailing portable optical-based methodologies in plant science and agronomy, and potentially before the detrimental impacts on yield become irreversible” to limited the comparison in optical-based sensing to eliminate any misleading here. Please see the modified sentence at Page 17 Line 338-340.